# The experience of urgent dialysis patients with end-stage renal disease: A qualitative study

**Min-Ling Lin[1], Kuei-Hui Chu[ID][2]***

1 MSN, RN, Head Nurse, Mackay Memorial Hospital, Taipei, Taiwan, 2 Department of Nursing, Ching Kuo Institute of Management and Health, Keelung, Taiwan

* kueihuichu@gmail.com

## Abstract

### Purpose

Taiwan is among the countries with the highest global prevalence of chronic renal disease. However, when advised to undergo dialysis therapy, patients with end-stage renal disease often hesitate. Attitudes toward medication and Taiwanese cultures are the main reasons for this delay, and delay conditioning requires urgent dialysis. This study aimed to explore the experience of urgent dialysis patients with end-stage renal disease.

### Methods

This study used a purposive sampling strategy with semi-structured interviews leading to in-depth interviews. Patients were recruited from the nephrology ward and hemodialysis center of a northern Taiwanese hospital. All participants were aged over 20 years with end-stage renal disease. Although advised by doctors to undergo dialysis, these patients delayed their treatment and later suffered severe complications. After emergency hospitalization, the patients' condition improved. Data were analyzed by content analysis.

### Results

Interviews with five participants suffering from end-stage renal disease identified six themes: "experiencing a sudden jolt," "silent organ brings the most pain," "feeling angry: why me?," "facing a dilemma," "taking risks," and "facing consequences."

### Conclusion

These patients delayed their treatment and later suffered severe complications, even though doctors advised them to undergo dialysis. Health professionals play an important role in communication and coordination, assisting patients in coping with their situation. The analysis of the reasons for the delay in undergoing dialysis, therefore, should help health professionals provide proper guidance and care to patients who are faced with the decision to accept dialysis treatment.

**Data Availability Statement:** All relevant data are within the paper and its Supporting information files.

**Funding:** Initials of the authors who received each award:Kuei Hui Chu Source of Funding This study

was funded by the Taiwan Nurses Association [grant no 1062002]. URL of each funder website: https://www.twna.org.tw/frontend/un10_open/welcome.asp The funders had no role in study design, data collection and analysis, decision to publish, or preparation of the manuscript.

**Competing interests:** The authors have declared that no competing interests exist.

## Introduction

In 2019, renal diseases, including nephritis, nephrotic syndrome, and nephropathy, were among the top 10 causes of mortality in Taiwan [1]. In 2016, the annual United States Renal Data System [2] reported that Taiwan had the highest reported prevalence rate of dialysis. Improper medication and the use of alternative treatments that are not thoroughly examined cause patients with end-stage renal disease (ESRD) to develop uremia within a short period of time [3]. When patients with ESRD are informed on the need for kidney transplantation or dialysis to improve quality of life, they tend to opt for alternative therapy [4].

In the early stages of chronic kidney disease (CKD), patients experience few signs or symptoms. Owing to the lack of knowledge about the disease and preference for folk or herbal medicine, patients with CKD can miss out on the benefits of appropriate early treatment. When it leads to irreversible renal failure (end-stage renal disease), only dialysis treatment can reduce the symptoms of the disease. If not treated by dialysis, patients may suffer from acute respiratory failure, metabolic acidosis, or other severe complications, delaying treatment and even causing death [5]. Some patients hesitate to act when faced with life-changing diseases [6–8]. Many patients experience uncertainty, fear, sadness, or anger [9] and refuse or delay starting dialysis [10] until they are forced to accept treatment after the development of life-threatening complications such as acute pulmonary edema and heart and respiratory failure [11]. This not only raises mortality rates but also causes worry for patients' families and increases medical costs [12–14].

Before the diagnosis of ESRD, most patients have preserved renal function. However, the feeling of helplessness when told of the need for dialysis and hesitancy to start dialysis often worsen their condition. When symptoms become life-threatening, patients will be arranged for urgent admission to the hospital for urgent dialysis. Therefore, this study aimed to explore the experience of urgent dialysis patients with end-stage renal disease.

## Methods

### Research design

This study used a qualitative design with purposive sampling to collect data. In-depth interview questions were designed in accordance with related literature, such as illness representations and coping processes of patients [4]. Data were collected between October and December 2016.

### Participants/Setting

The study recruited participants from the nephrology ward of a hospital in northern Taiwan. Participants were aged over 20 years and were diagnosed with ESRD accompanied by uremia. The characteristics of the study participants are summarized in Table 1. All five patients were diagnosed with ESRD. Although their doctors had recommended dialysis, these patients delayed treatment, resulting in severe complications (acute respiratory failure, acute pulmonary edema, hyperkalemia, cardiac arrhythmia, and heart failure) and admission to the intensive care unit. Later, these patients were hospitalized and underwent emergency hemodialysis. All the patients were conscious with no hearing difficulties and agreed to the interviews. Patients who had hemodialysis vascular access and were scheduled for treatment (AV-shunt and Hickman catheter) were excluded. The sample size was determined when data saturation was achieved. In other words, the researcher analyzed the information from the samples until no new information could be obtained.

Table 1. Characteristics of the study participants (N = 5).

|   | Sex | Age | Education Level | Marital Status | Work status | History | Diagnosis |
|---|-----|-----|-----------------|----------------|-------------|---------|-----------|
| A | Male | 62 | secondary | Widowed | Yes | Hypertension | ESRD |
| B | Male | 60 | Elementary school | Yes | Yes | Hypertension | ESRD |
| C | Male | 78 | Elementary school | No | No | DM, Hypertension | ESRD |
| D | Female | 72 | illiterate | Yes | No | Hypertension, DM, Heart disease | ESRD |
| E | Male | 78 | Elementary school | No | No | DM, Hypertension | ESRD |

## Data collection and analysis

The study was approved by the appropriate institutional review board (IRB-CGH-LP 105009). After explaining the purpose of the study to the participants, the researcher received written informed consent and made appointments for interviews. Interviews were conducted in environments where the participants felt most comfortable answering questions. The interviews were conducted privately, and participants were assured that their personal information would be kept confidential. The interviews were recorded and transcribed for further analysis and interpretation. All conversations took approximately 30 minutes.

## Instrument

A semi-structured interview guide was used for the in-depth interviews (Table 2). In line with the study's goal, the interviews aimed to explore the patients' mentality and understand the experience of urgent dialysis. The instruments used included the researcher's experiences, the interview guide, and a recorder to collect data during the interviews. Therefore, the researcher was the major instrument used in this study. After data collection and analysis, data was described consistently to meet the needs of dependability. Coding was conducted for data analysis and conceptualization.

## Results

From the results, six major themes were identified: "experiencing a sudden jolt," "silent organ brings the most pain," "feeling angry: why me?," "facing a dilemma," "taking risks," and "facing consequences." These are described below and summarized in Table 3.

## Experiencing a sudden jolt

At first, the patients did not experience any symptoms and were referred to the nephrology department from the emergency room. After interacting with the doctor, they became aware of their illness. Subsequently, feeling as though they had received a life sentence, they became

Table 2. Semi-structured interview guide used for the in-depth interviews.

| 1. Can you describe the disease process? |
|---|
| 2. What was the condition for this medical treatment? |
| 3. What kind of treatment method did the doctor suggest? |
| 4. How are you feeling right now? |
| 5. Do you accept the treatment method recommended by the doctor? (1) Yes (2) No (why?) |
| 6. What did you do when you declined the treatment? |
| 7. What is the reason for visiting the hospital this time? What symptoms do you have? |
| 8. Can you describe your feeling after the first dialysis experience? |

**Table 3. Six major themes identified in the study.**

| Themes |
| --- |
| Experiencing a sudden jolt |
| Silent organ brings the most pain |
| Feeling angry: why me? |
| Facing a dilemma |
| Taking risks |
| Facing consequences |

deeply frustrated. After the diagnosis, they sensed that their lives would completely change and felt sorry for themselves. Participant A: *"I was transferred to the nephrology department and underwent renal ultrasound, which showed renal atrophy."* Participant E: *"I had been suffering from leg cramps. So, it was recommended that I undergo a check-up at the hospital, where I was told that one of my kidneys was not functioning properly."*

## Silent organ brings the most pain

The kidneys are regarded as silent organs; there are no obvious symptoms of deteriorating function, causing the condition to be ignored. A diagnosis of kidney failure is made only when patients are aware of symptoms that consistently lower their quality of life. Participant A: *"Whenever the weather changed, I felt uncomfortable."* Participant C: *"I had edema and general discomfort, and I felt shortness of breath during work."*

## Feeling angry: Why me?

Participants who were recommended to undergo dialysis and prepare for vascular access felt furious and were unable to accept the fact, thereby attempting to avoid treatment. Participant B: *"I thought I should just procrastinate and keep working as dialysis would not cure my illness."* Participant D: *"I felt I would rather die than undergo dialysis."*

## Facing a dilemma

As kidney failure has no cure, participants hesitated when faced with the decision to undergo hemodialysis. Feeling uncomfortable, they could not make up their minds to accept the treatment. They felt like they were going back and forth on the decision and did not know where to go. In the interviews, participants expressed their sadness and disappointment. Participant A: *"I tried alternative treatment and religion. Neither was successful. I felt that if I accepted hemodialysis, it would take away my time and I would not be able to work."* Participant B: *"I asked around and most people said it would not help. I tried Chinese medicine and a different diet, but they were not effective. What was I to do? I did not want to undergo dialysis. I was worried and annoyed."* Participant C: *"I felt that if I underwent hemodialysis, I would have to spend a lot of time at the hospital, leaving me unable to do my job (farming). I needed to feed my wife and kids, but I was out of breath and unable to work. That was annoying."*

## Taking risks

While the participants hesitated to undergo dialysis, their condition became unbearable. At this stage, family, friends, and health professionals patiently supported and listened to them. Eventually, participants would start treatment. Participant B: *"I was worried that the burn wound on my left hand would be infected and I would end up having to undergo amputation, so*

*I finally agreed to dialysis. My family had always advised me to start dialysis.*" Participant D: "*My body has become like this. What could I do? I had no choice.*"

## Facing consequences

Without dialysis, participants suffered all kinds of pain, with their families experiencing mental stress. Participant B: "*I still do not want to do it. People die, stop pursuing me.*" Participant C: "*Dialysis goes on forever, and I will not be able to work. I would rather die.*"

## Discussion

This study aimed at exploring the experience of urgent dialysis patients with end-stage renal disease. Six themes were identified in this study, which included "experiencing a sudden jolt," "silent organ brings the most pain,", "feeling angry: why me?", "facing a dilemma," "taking risks," and "facing consequences." This result is similar to those of previous studies [14, 15]. However, among all the factors causing delay in dialysis therapy, this study identified facing a dilemma, which included family finances, to be the main reason. In the case of hemodialysis, patients are usually connected to a machine for three to four hours, three days a week, in a dialysis center and thus, their daily lives may be greatly affected [6, 13]. The participants experienced physical and mental challenges due to renal disease; all of them had experienced anxiety, sadness, denial, depression, low self-esteem, and helplessness at some point [16]. Therefore, they had trouble accepting their condition and refused or delayed dialysis therapy. Their reactions were consistent with those found in previous studies [14, 15].

Contextual information also provided findings similar to those of previous studies. Participants in this study sought alternative therapy when considering the possible side effects of dialysis and renal failure [8]. Another finding was that three out of five participants cited financial issues as the main reason for delaying dialysis therapy, which led to urgent dialysis. As these participants were the main breadwinners of their families, they were worried that undergoing dialysis therapy two to three times a week would interfere with their work, thereby affecting their earning potential. Therefore, it would be helpful if the government's social welfare department could issue disability cards and provide patients with access to social workers.

Patients and their families need an outlet to express their sorrow and anger. Considering these needs, health professionals must attempt to resolve their doubts and help them understand the treatment process. Previous research has demonstrated that the information presented to patients is often incomplete and difficult to grasp [17]. Providing accurate information and pre-dialysis education is essential in empowering patients with ESRD to choose dialysis modalities [6]. A video introduction to the dialysis process and environment could reduce patients' concerns. Health professionals could also refer patients to support groups and organizations that offer subsidies to help alleviate their economic burden. The goal should be to help patients cope with long-term dialysis therapy.

This study had a few limitations. First, the age range of the participants in this study was quite broad. As different age groups may face different kinds of stress, the results of this study are generalizable only to patients with ESRD from similar age groups. Therefore, future studies must recruit patients of varying ages. Another limitation was that the participants in this study were recruited from a specific regional hospital. Thus, it is not certain if the findings can be generalized to patients with ESRD from other regions.

During the interviews, the researcher tried to view the situation from the participants' perspective and understand their fear and hesitation with regard to dialysis therapy. From the interviews, it was evident that the patients needed companionship and someone to lend an ear. A trusting relationship with health professionals could help them accept proper treatment.

Health professionals should, therefore, be willing to provide mental support and companionship to patients with ESRD.

## Conclusion

Patients with ESRD have mixed feelings about dialysis therapy and focus on their negative emotions. Therefore, health professionals should play an important role by communicating and coordinating with patients to help them cope with their situation, and make them feel confident to accept proper treatment.

## Supporting information

**S1 File.**
(PDF)

## Author Contributions

**Conceptualization:** Min-Ling Lin.

**Data curation:** Kuei-Hui Chu.

**Formal analysis:** Kuei-Hui Chu.

**Methodology:** Kuei-Hui Chu.

**Writing – original draft:** Min-Ling Lin, Kuei-Hui Chu.

**Writing – review & editing:** Min-Ling Lin, Kuei-Hui Chu.

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
