## [Decision Letter · Decision Letter 0]

7 May 2021

PONE-D-21-04435

The experience of urgent dialysis patients with end-stage renal disease: a qualitative study

PLOS ONE

Dear Dr. CHU,

Thank you for submitting your manuscript to PLOS ONE. After careful consideration, we feel that it has merit but does not fully meet PLOS ONE’s publication criteria as it currently stands. Therefore, we invite you to submit a revised version of the manuscript that addresses the points raised during the review process.

We look forward to receiving your revised manuscript.

Kind regards,

Tareq Mukattash

Academic Editor

PLOS ONE

Journal Requirements:

2)  Please ensure that you include a title page within your main document. We do appreciate that you have a title page document uploaded as a separate file, however, as per our author guidelines (http://journals.plos.org/plosone/s/submission-guidelines#loc-title-page) we do require this to be part of the manuscript file itself and not uploaded separately.

3) Thank you for including your ethics statement:  "Institutional Review Board of the Cathay General Hospital

he study was approved by the appropriate institutional review board (IRB-CGH-LP 105009). "

Please amend your current ethics statement to confirm that your named institutional review board or ethics committee specifically approved this study.

4) PLOS requires an ORCID iD for the corresponding author in Editorial Manager on papers submitted after December 6th, 2016. Please ensure that you have an ORCID iD and that it is validated in Editorial Manager. To do this, go to ‘Update my Information’ (in the upper left-hand corner of the main menu), and click on the Fetch/Validate link next to the ORCID field. This will take you to the ORCID site and allow you to create a new iD or authenticate a pre-existing iD in Editorial Manager. Please see the following video for instructions on linking an ORCID iD to your Editorial Manager account: https://www.youtube.com/watch?v=_xcclfuvtxQ

Reviewers' comments:

Reviewer's Responses to Questions

**Comments to the Author**

1. Is the manuscript technically sound, and do the data support the conclusions?

Reviewer #1: Partly

2. Has the statistical analysis been performed appropriately and rigorously? 

Reviewer #1: N/A

3. Have the authors made all data underlying the findings in their manuscript fully available?

Reviewer #1: No

4. Is the manuscript presented in an intelligible fashion and written in standard English?

Reviewer #1: Yes

5. Review Comments to the Author

Reviewer #1: Dear Authors

Thanks for the great efforts made to accomplish this study. However, the manuscript need further work before it could be published. The major fall of the manuscript is mainly related to the method and finding analysis. I included all my comments directedly to the attached comments for ease of getting comments. I did few changes to the sentences to improve how it read and highlighted them in yellow for you. Good luck

6. PLOS authors have the option to publish the peer review history of their article (what does this mean?). If published, this will include your full peer review and any attached files.

Reviewer #1: **Yes: **Dr. Manal Kassab

---

## [Author Response · Author response to Decision Letter 0]

1 Aug 2021

We have modified the manuscript, which is now entitled “The experience of urgent dialysis patients with end-stage renal disease: a qualitative study” (PONE-D-21-04435), according to your comments. Our revisions are highlighted in red in the manuscript, and our modifications are detailed below. Thank you very much for reviewing our manuscript. We look forward to hearing from you.

Sincerely,

---

## [Decision Letter · Decision Letter 1]

15 Dec 2021

The experience of urgent dialysis patients with end-stage renal disease: a qualitative study

PONE-D-21-04435R1

Dear Dr. CHU,

We’re pleased to inform you that your manuscript has been judged scientifically suitable for publication and will be formally accepted for publication once it meets all outstanding technical requirements.

Kind regards,

Tareq Mukattash

Academic Editor

PLOS ONE

Additional Editor Comments (optional):

Reviewers' comments:

Reviewer's Responses to Questions

**Comments to the Author**

1. If the authors have adequately addressed your comments raised in a previous round of review and you feel that this manuscript is now acceptable for publication, you may indicate that here to bypass the “Comments to the Author” section, enter your conflict of interest statement in the “Confidential to Editor” section, and submit your "Accept" recommendation.

Reviewer #1: All comments have been addressed

2. Is the manuscript technically sound, and do the data support the conclusions?

Reviewer #1: Yes

3. Has the statistical analysis been performed appropriately and rigorously? 

Reviewer #1: Yes

4. Have the authors made all data underlying the findings in their manuscript fully available?

Reviewer #1: Yes

5. Is the manuscript presented in an intelligible fashion and written in standard English?

Reviewer #1: Yes

6. Review Comments to the Author

Reviewer #1: The manuscript looks much better and all required modifications have been made. Minor editing is needed only

7. PLOS authors have the option to publish the peer review history of their article (what does this mean?). If published, this will include your full peer review and any attached files.

Reviewer #1: No

---

## [Editor Report · Acceptance letter]

20 Dec 2021

PONE-D-21-04435R1 

The experience of urgent dialysis patients with end-stage renal disease: a qualitative study 

Dear Dr. CHU:

I'm pleased to inform you that your manuscript has been deemed suitable for publication in PLOS ONE. Congratulations! Your manuscript is now with our production department. 

Kind regards, 

on behalf of

Dr. Tareq Mukattash 

Academic Editor

PLOS ONE